# Vector Road Map Updating from High-Resolution Remote-Sensing Images with the Guidance of Road Intersection Change Detection and Directed Road Tracing

Haigang Sui [1], Ning Zhou [1], Mingting Zhou [1,*] and Liang Ge [2]

1 State Key Laboratory of Information Engineering in Surveying, Mapping and Remote Sensing, Wuhan University, Wuhan 430072, China
2 Tianjin Institute of Surveying and Mapping Company Limited, No. 9 Changling Road, Liqizhuang, Tianjin 300060, China
* Correspondence: mintyzhou@whu.edu.cn

**Abstract:** Updating vector road maps from current remote-sensing images provides fundamental data for applications, such as smart transportation and autonomous driving. Updating historical road vector maps involves verifying unchanged roads, extracting newly built roads, and removing disappeared roads. Prior work extracted roads from a current remote-sensing image to build a new road vector map, yielding inaccurate results and redundant processing procedures. In this paper, we argue that changes in roads are closely related to changes in road intersections. Hence, a novel changed road-intersection-guided vector road map updating framework (VecRoadUpd) is proposed to update road vector maps with high efficiency and accuracy. Road-intersection changes include the detection of newly built or disappeared road junctions and the discovery of road branch changes at each road junction. A CNN-based intersection-detection network (CINet) is adopted to extract road intersections from a current image and an old road vector map to discover newly built or disappeared road junctions. A road branch detection network (RoadBranchNet) is used to detect the direction of road branches for each road junction to find road branch changes. Based on the discovery of direction-changed road branches, the VecRoadUpd framework extracts newly built roads and removes disappeared roads through directed road tracing, thus, updating the whole road vector map. Extensive experiments conducted on the public MUNO21 dataset demonstrate that the proposed VecRoadUpd framework exceeds the comparative methods by 11.01% in pixel-level Qual-improvement and 13.85% in graph-level F1-score.

**Keywords:** vector road map update; road intersection change detection; directed road tracing; high-resolution remote-sensing images

## 1. Introduction

Highly updated road maps are crucial in applications, such as intelligent transportation, autonomous driving, and disaster emergency response. High-resolution remote-sensing imagery with wide coverage and fast update speeds has been the main data source to update road maps collected historically. Most of the current research focuses on extracting vector road maps from remote-sensing images [1–3].

Given that there are already high-quality road maps, such as the Open Street Map, covering the world, the construction of vector road maps has gradually transitioned from road extraction from scratch to updating changed roads [4,5]. However, even with high-quality road maps as the basis, updating a vector road map is still a labor-intensive task [4,6,7]. Hence, there is an urgent need to develop automatic updating methods for vector road maps based on high-resolution remote-sensing images. Road centerline extraction methods from remote-sensing images are used to update vector road maps.

With the increasing availability of high-resolution remote-sensing images, many road centerline extraction methods based on high-resolution images have been proposed in the past decades [8–10]. These methods can be divided into methods based on road segmentation, methods based on direct graph extraction, and multi-task methods that extract both the road surface and the centerline. Road segmentation-based methods first segment the road surface and then obtain the road centerline by thinning the road surface [11–17].

However, the road surface segmentation itself has many difficulties, and the thinning process is prone to producing centerline disconnections and burrs. To improve the topological connectivity of road centerlines, methods based on direct graph extraction were proposed to directly infer road maps from different viewpoints [18,19]. In addition, some multi-task cascade networks are proposed to extract road surface and road centerline simultaneously [20–22].

The current methods obtain high accuracy in road centerline extraction. However, complex post-processing should be conducted on the discontinuous and burred road centerline extraction results to add missed roads, connect broken sections, and remove false roads when using these results to update a historical road map. Therefore, it is necessary to perform research on road map updating based on remote-sensing images directly.

Road map updating involves verifying unchanged roads, extracting newly built roads, and removing disappeared roads. Taking the public MUNO21 dataset [4] as a starting point, there has been some research on updating roads based on change detection in bi-temporal remote-sensing images. For example, Bastani et al. [23] proposed a two-stage road update framework based on bi-temporal imagery. Zhou et al. [5] proposed the UGRoadUpd framework for guiding road updates by unchanged roads. However, obtaining a historical remote-sensing image that matches the collection time of the historical road map is difficult. Therefore, updating a historical road map with only a current image remains an issue.

To solve the above-mentioned problem, a novel road vector map updating (VecRoad-Upd) framework is proposed based on the observation that road changes are highly correlated with road-intersection changes. Road-intersection changes involve changes to the locations and the branches of the intersections. To accurately discover the change in road intersections, a CNN-based intersection-detection network (CINet) is used to extract intersections from current images and historical vector road maps.

A metric named the Threshold of Partial Intersection-of-Union ($T_{PIoU}$) is introduced to measure whether there are newly built or disappeared road junctions. Based on the discovery of changed intersections, a road branch detection network (RoadBranchNet) and a spatial analysis strategy are combined to detect intersection branches in current images and old road maps. A threshold of angle ($T_{angle}$) is used to assess whether the direction of road branches changed for a homonymic road intersection. Based on the discovery of direction-changed road branches, directed road tracing is designed to update road maps accurately.

The remainder of this paper is organized as follows: Section 2 introduces the related road extraction and road map update methods. Section 3 introduces an overview of the proposed road change detection and update framework. Section 4 presents the experimental results and analysis. Section 5 demonstrates the ablation analyses. Section 6 shows and explains the failure cases. Our conclusions are presented in Section 7.

## 2. Related Work

### 2.1. Road Extraction

#### 2.1.1. Road Surface Segmentation

The road surface segmentation methods based on remote-sensing images are mainly divided into traditional methods and deep-learning-based methods. In traditional methods, the road surface is segmented by manually designing features [24,25] and combining theories about statistics and machine learning, such as support vector machine (SVM) [26,27], artificial neural network (ANN) [28,29], and maximum likelihood [30].

However, the shallow features used in these traditional methods are usually suitable for small areas only, and these traditional methods usually yield more missed detections when dealing with satellite images of large areas. With the development of deep learning, Mnih et al. [31] presented the first work that used deep neural networks to detect road networks in aerial images. They first segmented the image into small chunks, then predicted the road network within each chunk, and finally merged the chunks to obtain the final road surface segmentation map. Most of the subsequent road surface segmentation methods [32–34] continue a similar approach but use more effective segmentation networks, such as U-Net [35], DeepLab V3+ [36], and SegNet [37].

These methods usually obtain completed road segmentation results; however, they do not guarantee road connectivity. To improve the connectivity of road segmentation results, Batra et al. [12] proposed a stacked multi-branching module that can effectively use the association information between road segmentation and directed learning tasks to improve road connectivity. Mei et al. [38] proposed a connectivity attention module and designed CoANet to explore the relationship between neighboring pixels in an image to deal with road breakage due to the occlusion of trees, shadows, etc.

Compared with the traditional methods, the deep-learning method benefits from its powerful feature-extraction capability to extract rich road semantic information from remote-sensing images and obtain higher accuracy road surface segmentation results. However, the existing road surface segmentation methods have difficulty constructing a complete road topology. Therefore, road centerline extraction methods aiming at building a complete road topology are gradually derived.

### 2.1.2. Road Centerline Extraction

Automatically inferring road centerlines from remote-sensing imagery is a well-studied subject. Many road centerline extraction methods have been proposed in the past decades [2,31,39–46]. These methods are mainly divided into methods based on road segmentation, methods based on direct graph extraction, and multi-task methods that extract both the road surface and centerline. Road segmentation-based methods first segment the road surface and then obtain the road centerline by thinning the road surface [11–17].

Zhu et al. [16] extracted the road surface based on the gray morphological characteristics, and then extracted the road centerline by the line segment match method. Liu et al. [17] first extracted the road surface by CNN and then extracted road centerlines using multiscale Gabor filters and multiple directional non-maximum suppression. However, extracting centerlines from road segmentation requires complex post-processing and can be influenced by inaccurate road segmentation results, leading to disconnected centerline topologies.

Unlike segmentation-based methods, the graph-extraction approach learns the graph structure directly to improve the road map connectivity [2,3,18,45–49]. For example, Bastani et al. [45] proposed an iterative road centerline tracing method called RoadTracer. RoadTracer generates a window centered on the current location at each step of the tracing to determine the direction and action of the next tracing step. Limited by the number of starting points, locations, and fixed step lengths, the road network extracted by RoadTracer often leads to incompleteness and road offset at intersections.

To improve completeness, Wei et al. [47,48] proposed the multiple starting point tracing strategy (MspTracer). MspTracer traces the road centerline using multiple intersections in the road segmentation as starting points. Finally, the road segmentation results and the road centerline are fused to obtain a more complete and connected road network. To correct the road offset due to the fixed step length in RoadTracer, Tan et al. [3] proposed VecRoad with adaptive step length and segmentation guidance. VecRoad obtains a more accurate road map by uniformly constraining the tracking direction and step length in each step. Although iterative road tracing can maintain road connectivity well, it is time-consuming.

Therefore, He et al. [46] proposed a unified framework for generating road graphs directly from images (Sat2Graph). The framework encodes the road graph as a tensor through graph tensor encoding (GTE) to train a simple, non-recursive, supervised model.

The model predicts the road graph as a whole from the input image and achieves a complete road extraction result. To improve the efficiency of road map extraction and further enhance the completeness of the road map, Gaetan et al. [49] proposed a method to directly infer the final road map in a single pass.

In addition, to utilize the symbiotic relationships between the road surface and centerline to enhance the road extraction integrity and connectivity, some multi-task cascade networks have been proposed [20–22,50,51]. For example, Cheng et al. [20] proposed a cascaded end-to-end CNN (CasNet) to simultaneously process road segmentation and centerline extraction tasks for very high-resolution (VHR) remote-sensing images. Liu et al. [50] developed a multi-task cascaded CNN called RoadNet to simultaneously predict the road surface, centerline, and boundary, which was the first attempt to unify the three road extraction tasks.

A framework for the cascading prediction of the road surface, centerline, and boundary (CasMT) was similarly proposed by Lu et al. [51]. Topology-aware learning was applied in this framework to capture the road topology and focus on hard samples using hard-sample mining loss (HEM) to further enhance road integrity. Existing technologies have greatly improved the accuracy of road centerline extraction. However, factors, such as road material changes as well as tree and building shading can still affect the quality of the road centerline network. This is limited by the complex post-processing steps required to apply inaccurate road centerline networks to update historical vector road maps. Therefore, it is necessary to conduct research to update the road maps directly based on remote-sensing images.

### 2.2. Road Map Update

The key to road map updating is to verify unchanged roads, extract newly built roads, and remove disappeared roads instead of extracting the road map from scratch. In past studies, researchers focused more on updating vector road maps based on vehicle GPS [52,53]. However, the newly built roads added by these map update methods showed false-positive errors due to GPS noise. Furthermore, the coverage of GPS tracks is lower than that of satellite imagery; therefore, in this paper, we focus on using satellite imagery for road updates since it is globally available.

In recent years, the increasing availability of high-resolution remote-sensing imagery has sparked interest in road map updating by processing remote-sensing images [4,5,23,54]. For example, Wei et al. [54] proposed a road update strategy based on road segmentation and historical road maps; however, the strategy was limited by the accuracy of the road segmentation results. Bastani et al. [4] extended the existing state-of-the-art road extraction method for road updating on the road updating dataset (MUNO21).

However, limited by the accuracy of the road extraction results, inaccurate road update results are exhibited. Therefore, some road update methods based on bi-temporal remote-sensing image change detection networks [55–57] have been proposed in recent years. For example, Bastani et al. [23] proposed a two-stage road update framework from the perspective of change detection based on bi-temporal imagery. The first stage uses iterative road tracing to find candidate changed roads; the second stage uses self-supervised change detection to filter them; and finally, the framework updates the road map accurately.

Zhou et al. [5] proposed the UGRoadUpd framework for guiding road updates by unchanged roads. This framework improves the quality of the updated road network by limiting the road update range and learning features from unchanged roads. However, both methods above are based on the change detection of bi-temporal images to discover changed roads, which requires a high temporal match between historical images and road maps. Furthermore, obtaining historical remote-sensing images with the time match of historical road maps is typically difficult.

Therefore, updating a historical road map with only a new-temporal image remains an issue. In this paper, a novel vector road map updating (VecRoadUpd) framework guided by changed intersections is proposed to update historical vector road maps. Intersection change detection is conducted directly on current images and historical road maps, and

directed tracing is used to limit the direction of road tracking to improve the efficiency and accuracy of the road updates.

## 3. Methodology

The vector road map updating (VecRoadUpd) framework proposed in this paper updates road maps by detecting changed road intersections and tracing roads directionally. Different from the existing road map updating methods, VecRoadUpd captures possible road changes by discovering road-intersection changes. Taking the location and direction of changed road branches, VecRoadUpd updates road maps accurately through directed road tracing. The workflow of the VecRoadUpd framework is shown in Figure 1.

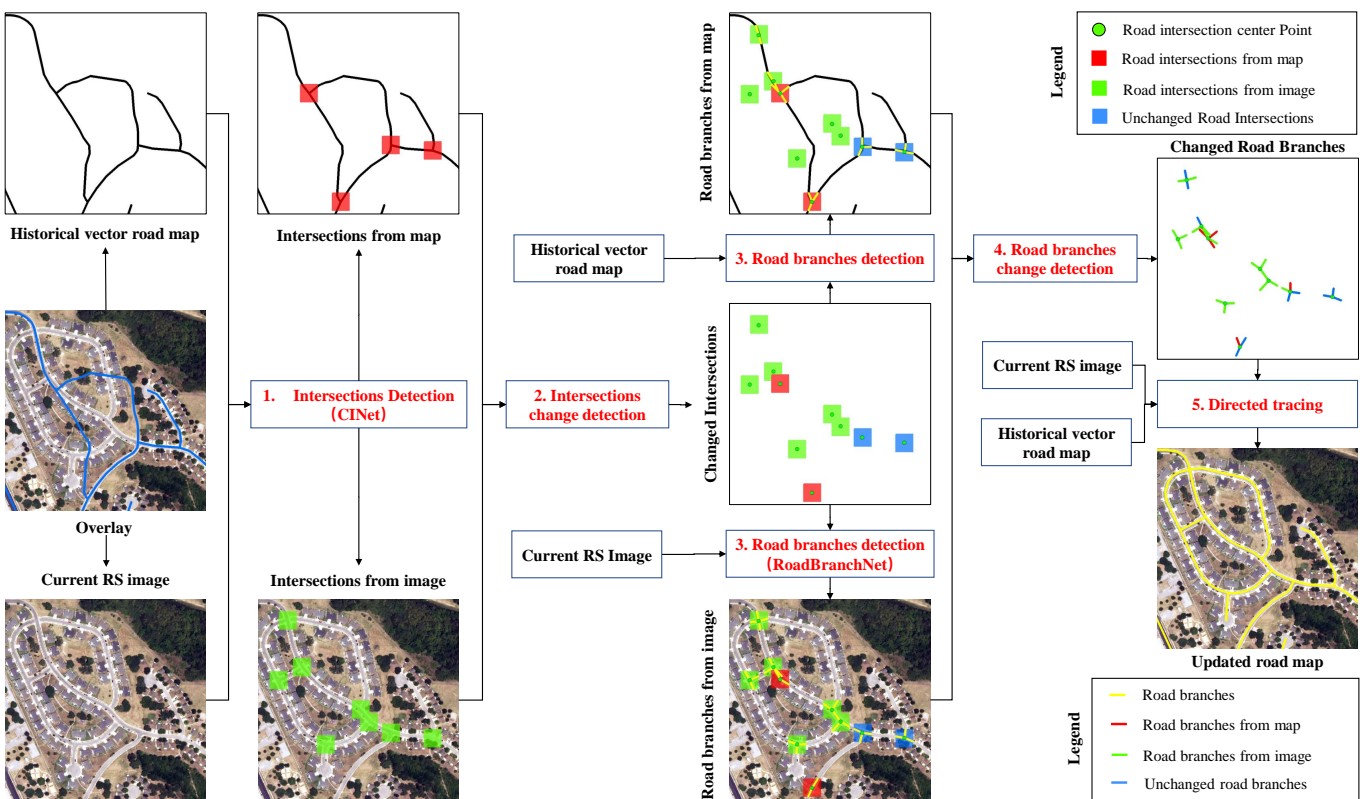

**Figure 1.** Flowchart of the proposed vector road map updating (VecRoadUpd) framework.

It can be seen from Figure 1 that the proposed VecRoadUpd framework takes a current remote-sensing image and an old road vector map as input and directly outputs an updated road vector map. The VecRoadUpd framework includes road intersection change detection, road branch change detection, and directed road tracing. The intersection change detection process finds newly built and disappeared intersections. Newly built intersections mean there are newly built roads added, and disappeared intersections mean there are disappeared roads.

Newly built and disappeared roads can also be built and removed from existing road intersections, so VecRoadUpd extracts the road branches of intersections by using a road branch detection network (RoadBranchNet) in the second stage. In this way, VecRoadUpd can detect changes in the old road map while also providing changed road branch directions for tracing newly built roads and removing disappeared roads. Based on the changed road branch directions, VecRoadUpd updates road maps through directed road tracing in the last stage.

### 3.1. Road Intersection Change Detection

Road intersection change detection is used to find newly built and disappeared intersections. The idea of extracting the new and old temporal road intersections first and

change detection later is used to discover the changed road intersections. To extract the new and old temporal road intersections, a CNN-based intersection-detection network (CINet) is applied to extract road intersections from a current remote-sensing image and a historical road vector map. The CINet uses CSPDarkNet53 [58] as the backbone, FPN [59] as the feature fusion neck, and a decoupled head [60] commonly used in one-stage object-detection networks as the head.

The architecture of CINet is shown in Figure A1. A hybrid loss function consisting of an object confidence loss ($l_{obj}$), a classification loss ($l_{cls}$), and a target box regression loss ($l_{iou}$) is used to train CINet to learn features, such as the shape and texture of intersections. Among them, $l_{obj}$ and $l_{cls}$ are calculated using binary cross-entropy loss ($l_{BCE}$), and $l_{iou}$ is calculated using Complete IoU (CIoU) loss [61]. A well-trained CINet is used to extract road intersections from current remote-sensing images and historical road vector maps.

$$l_{BCE} = \frac{1}{N}\sum_{i=0}^{N}(Y(i)_{gt}log(Y(i)_{pred}) + (1 - Y(i)_{gt})log(1 - Y(i)_{pred})) \tag{1}$$

where $Y(i)_{gt}$ represents the true value and $Y(i)_{pred}$ represents the predicted value.

Then, an intersection change analysis rule is presented to find newly built and disappeared road junctions based on the road intersection extraction results. A novel indicator named partial intersection over union (PIoU) is developed to judge whether two boxes belong to the same intersection. The PIoU is calculated by the formula PIoU = max (area $(A \cap B)$/area $(A)$, area $(A \cap B)$/area $(B)$). Based on PIoU, the road intersection change analysis rule is introduced to detect newly built and disappeared intersections as shown in Figure 2.

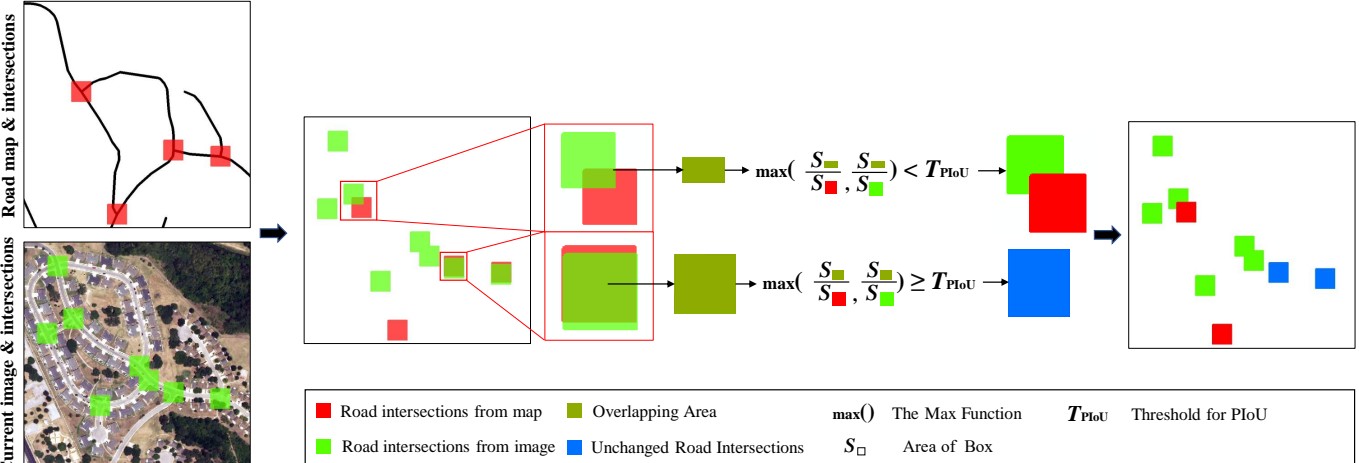

**Figure 2.** The process of road intersection change detection.

The road intersection change analysis rule takes new and old temporal intersections extracted from a current image and a historical road map as inputs and outputs the road intersection change detection result. In this rule, overlay analysis is first performed to obtain the overlap area between the old and new intersections, and then the PIoU is calculated. If the calculated PIoU value is less than the threshold of PIoU ($T_{PIoU}$), the intersection is considered a changed intersection and, otherwise, is considered unchanged.

It is worth noting that the value of $T_{PIoU}$ is adjustable and is a variable that affects the intersection change detection and road map update. Therefore, the detailed experiments about how $T_{PIoU}$ influences the road update accuracy were conducted in Section 5.1. Through intersection change detection, changes in the historical road map are initially detected. Furthermore, the prior information required for road branch change detection is also obtained: the position of the intersection and the number ($N$) of road branches connected to the intersection.

### 3.2. Road Branch Change Detection

The intersection change detection in Section 3.1 obtains the location of the changed intersection and the number (*N*) of road branches connected to the intersection without obtaining specific changed road branches. Therefore, to obtain more specific changed road branches, the road branch change detection process is designed in Section 3.2. The road branch change detection process consists of two steps, road branch extraction and change discovery, based on the changed intersections obtained in Section 3.1.

#### 3.2.1. Road Branch Detection

Road branch detection aims to extract road branch directions from current remote-sensing images and historical vector road maps. Due to the different data types of images and vector road maps, different road branch detection methods are used in this paper. The flowchart of the two road branch detection methods is shown in Figure 3.

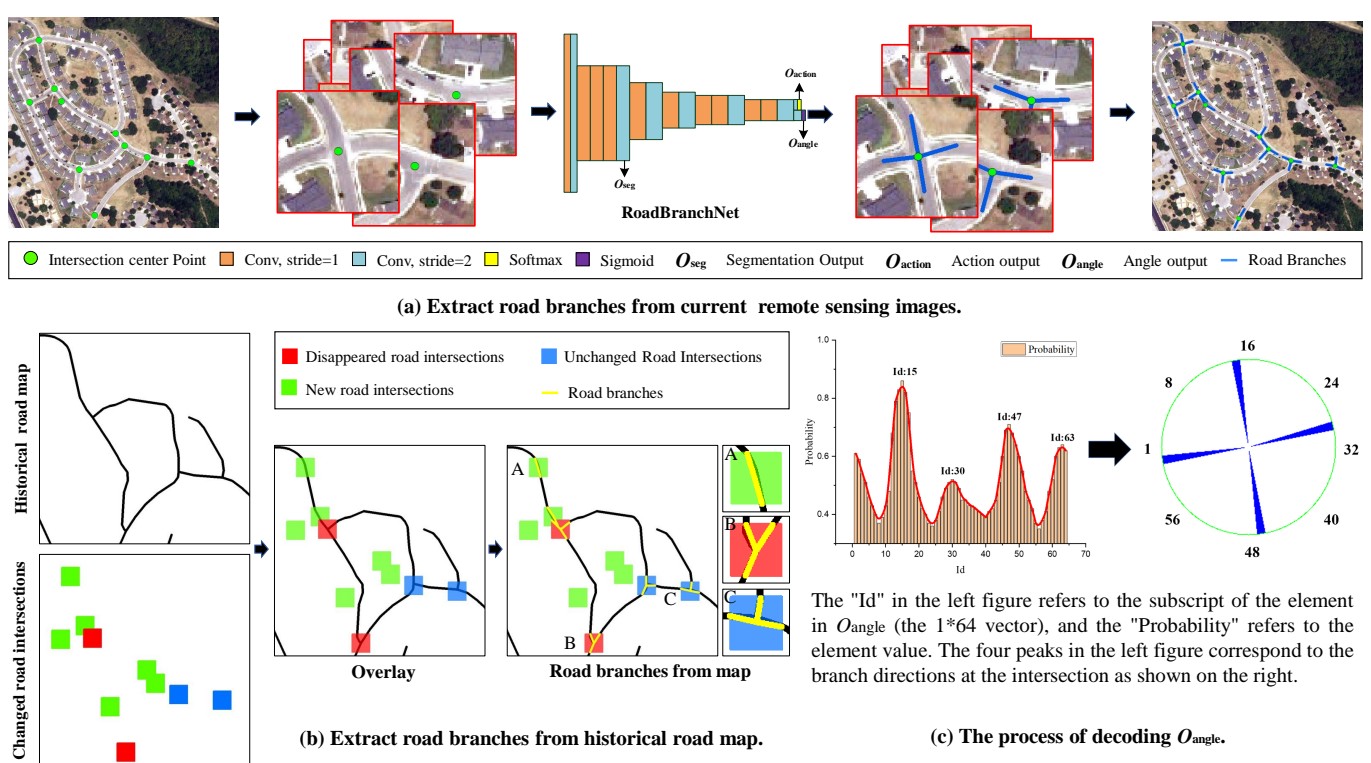

**(a) Extract road branches from current remote sensing images.**

**(b) Extract road branches from historical road map.**

The "Id" in the left figure refers to the subscript of the element in $O_{angle}$ (the 1*64 vector), and the "Probability" refers to the element value. The four peaks in the left figure correspond to the branch directions at the intersection as shown on the right.

**(c) The process of decoding $O_{angle}$.**

**Figure 3.** Extract road branches from current remote-sensing images and historical vector road maps.

Figure 3 illustrates the detailed process of extracting road branches from a current image and a historical vector road map. As seen in Figure 3a, a road branch detection network (RoadBranchNet) is used to extract road branch directions from the current remote-sensing image. The RoadBranchNet is inspired by the CNN-based decision module designed in RoadTracer. However, the decision module in RoadTracer decodes only one road branch direction from the angle output ($O_{angle}$) and cannot obtain multiple road branch directions at intersections.

Therefore, we modified the output layer of the decision module and decoded $O_{angle}$ using local maximum analysis. The structure of RoadBranchNet is shown in Figure 3a. The process of decoding $O_{angle}$ is shown in Figure 3c. The $O_{angle}$ is a $1 \times 64$ vector, where "64" means that the $2\pi$ radian centered on the intersection is divided into 64 equal parts, and each part represents a branch direction. Each value in the $O_{angle}$ represents the probability that the intersection has branches in each direction.

To obtain road branch directions, the local maximum analysis is conducted on these 64 probability values. Furthermore, the local maximums are sorted by the number (*N*) of road branches connected to the intersection. Then, the directions represented by the larger

*N* probability local maximums are the branch directions of the intersection. In this way, the road branches for different intersections are extracted from a current remote-sensing image.

As shown in Figure 3b, a road branch detection process based on the spatial analysis strategy is used to extract road branch directions from a historical vector road map. First, the spatial analysis strategy of the road intersection box and vector road map is performed to obtain the cross points. Then, the road intersection center point is connected with the corresponding cross points to obtain the road branch directions. In this way, the road branches for different intersections are extracted from a historical vector road map.

### 3.2.2. Road Branch Change Detection

A branch change detection process based on the intersection branch detection results is presented in this section to find newly built, disappeared, and unchanged road branches in old road maps from current images. The main idea is to obtain changed and unchanged road branches by comparing the differences between the directions of corresponding old and new road intersection branches detected from old road maps and new remote-sensing images. The workflow is shown in Figure 4.

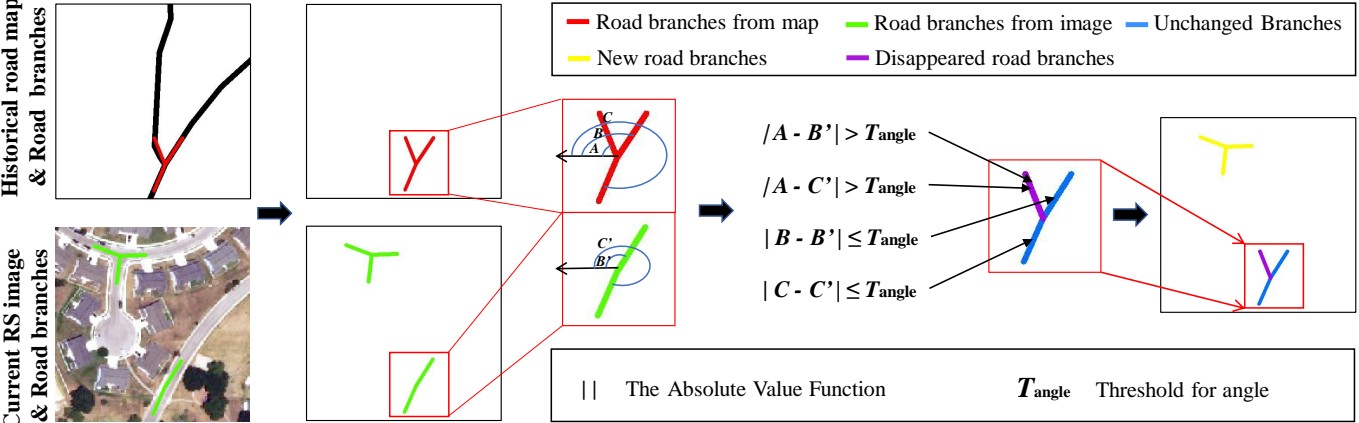

**Figure 4.** Flowchart of road branch change detection.

The road branch change detection is performed by comparing the angles between the road branches detected from a current image and a vector road map. As shown in Figure 4, the road branch direction is calculated with the intersection center point as the origin and the horizontal left direction as the positive direction. Then, the absolute difference between road branches detected from a current image and the vector road map is calculated. If the difference is less than or equal to $T_{angle}$, the two branches are considered to be the same branch; otherwise, the newly added or disappeared branches are obtained. $T_{angle}$ is set to $\pi/8$, and the influence of the value setting of $T_{angle}$ on the road map update is analyzed in Section 5.2.

### 3.3. Directed Road Tracing

A directed road-tracing strategy inspired by RoadTracer was applied to extract newly built roads and verify changed roads in the proposed road map updating process. In RoadTracer, however, the tracing starting points need to be given manually, which reduces the degree of automation of the algorithm. Although Wei et al. [47] used multiple starting point tracing (MspTracer) to improve the automation of RoadTracer, MspTracer cannot extract a complete road map because the starting points of MspTracer rely on incomplete road segmentation results. Different from the above-mentioned road tracing methods, our directed road-tracing strategy takes the intersections extracted from images and vector road maps as the starting points and uses changed road branch directions as the initial tracing directions to extract and verify the changed roads.

As shown in Figure 5, directed road tracing takes the road branches, the old road map, and the current image as input and outputs the updated road map.

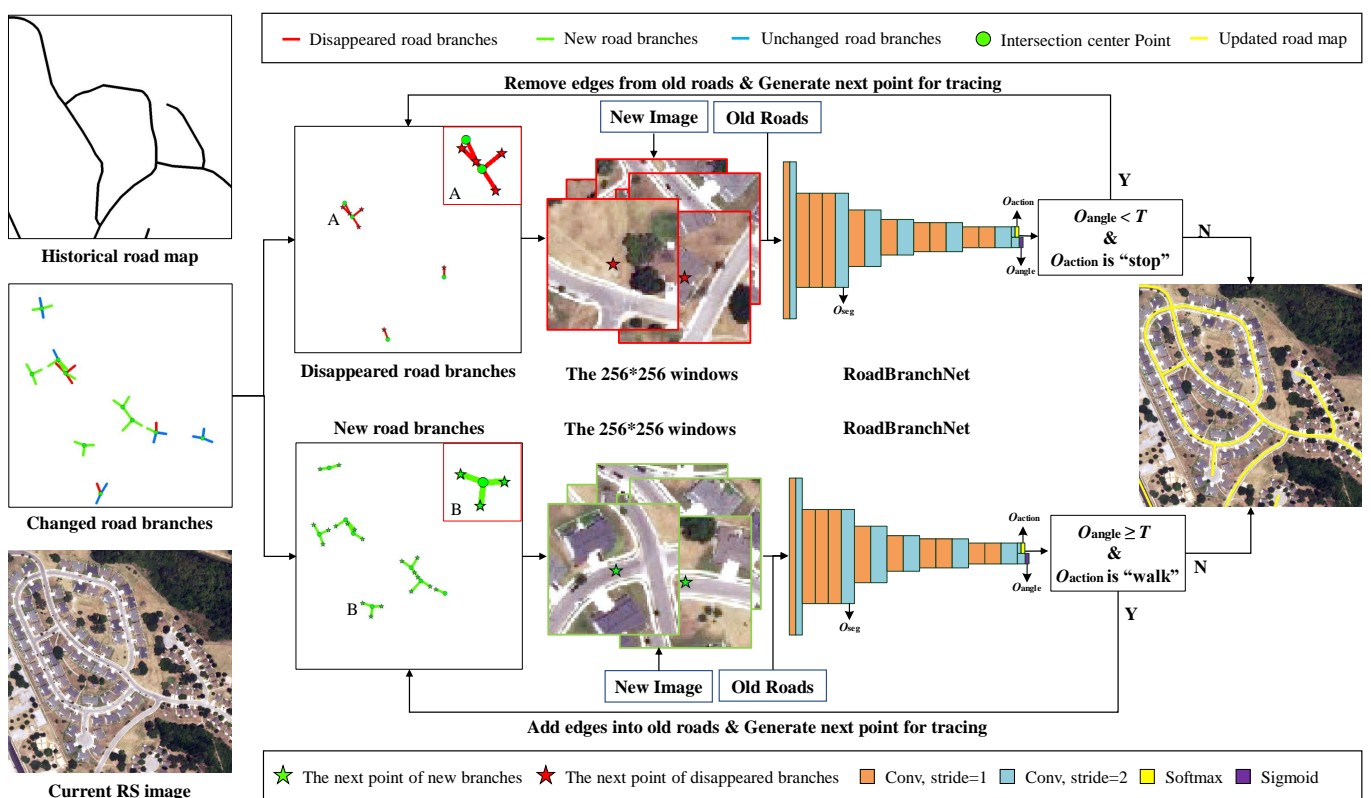

**Figure 5.** Flowchart of directed tracing for road map updates.

It can be seen from Figure 5 that the directed road tracing is divided into two processing steps based on the class of direction-changed branches. For new branches, directed tracing aims to extract the newly built roads that do not exist in old road maps but exist in current images. First, a starting point stack and an initial direction stack are generated from intersection center points and new branch directions, and $256 \times 256$ windows are generated with the image as the base map and the starting points as the center. These windows are then fed into the CNN decision module, and the output of this module decides whether to continue tracing. If continuing tracing, it repeats the above steps; otherwise, it returns to the current starting point or direction from the stack and starts tracing from the next starting point or direction.

We consider the computational cost of tracing duplicate roads. We stipulate that, if the intersection center points are tracked, the tracing will stop, and the current starting point will be popped from the stack to start tracing from the next starting point until the stack is empty. For disappeared branches, directed tracing aims to remove and validate vanishing roads that exist in old road maps but not in current images. The processing steps for disappeared branches are similar to those for new branches with the main difference being how the CNN-based decision module output is handled. For new branches, when the angle output ($O_{angle}$) is greater than or equal to the tracing threshold ($T$) and the action output ($O_{action}$) is "*walk*", the algorithm will add the tracked edges to the current road map and generate a next point to continue tracing; otherwise, it will stop tracing.

For disappeared branches, when $O_{angle}$ is smaller than $T$ and $O_{action}$ is "*stop*", the algorithm will remove the edges that have been verified as disappeared from the historical road map and generate a next point to continue the verification; otherwise, the verification will be stopped. In this way, when both the starting points stack and the initial directions stack are empty, the updated road map is obtained.

## 4. Experiments

### 4.1. Experimental Setups

#### 4.1.1. Dataset Description

To evaluate the effectiveness of our proposed VecRoadUpd, we conducted extensive experiments on the public road map update dataset named MUNO21. MUNO21 is a large-scale dataset for vector road map updating that includes pairs of road maps and remote-sensing imagery. The road maps are from OpenStreetMap (OSM), and the imagery is from the National Agriculture Imagery Program (NAIP), covering a total of 21 cities in the US with a total area of 6052 square kilometers. The core part of this dataset is a set of 514 map update scenes and 780 no-change scenes. Each scene contains bounding boxes (x, y, w, and h), a pre-change map G, and a post-change map G*.

The entire dataset is divided into a training set containing 10 cities and 726 scenes and a test set containing 11 cities and 568 scenes. Each scene is labeled with one or more tags (e.g., Constructed, Was-missing, Deconstructed, Was-incorrect, and No-change), which can be easily used to update road maps and evaluate using this dataset. To extract road intersections from the imagery, a road intersection dataset (WuHan Road Intersection, WHRI) was manually annotated. The source images include Google Earth images from Wuhan and binary maps converted by OSM. An illustration of the WHRI dataset is shown in Figure 6.

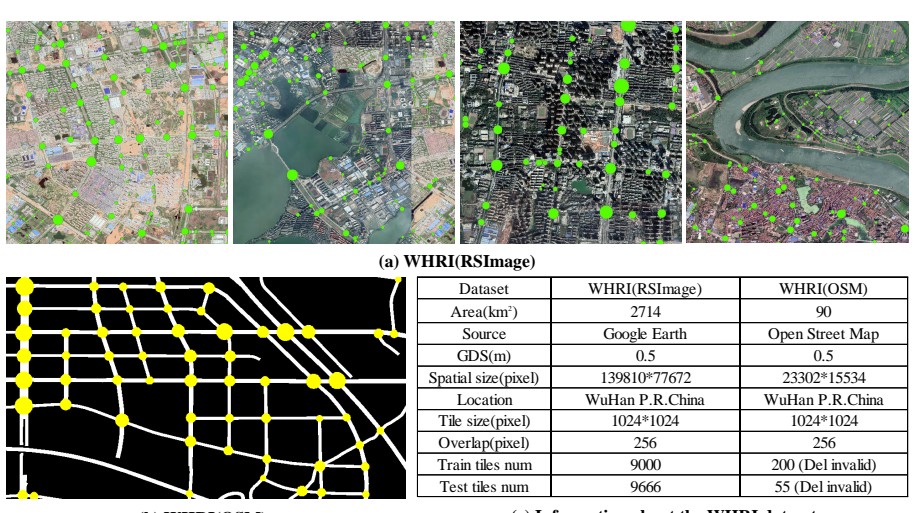

**(a) WHRI(RSImage)**

**(b) WHRI(OSM)**

| Dataset | WHRI(RSImage) | WHRI(OSM) |
|---|---|---|
| Area(km²) | 2714 | 90 |
| Source | Google Earth | Open Street Map |
| GDS(m) | 0.5 | 0.5 |
| Spatial size(pixel) | 139810*77672 | 23302*15534 |
| Location | WuHan P.R.China | WuHan P.R.China |
| Tile size(pixel) | 1024*1024 | 1024*1024 |
| Overlap(pixel) | 256 | 256 |
| Train tiles num | 9000 | 200 (Del invalid) |
| Test tiles num | 9666 | 55 (Del invalid) |

**(c) Information about the WHRI dataset**

**Figure 6.** Illustration of the WHRI dataset.

#### 4.1.2. Implementation Details

In this paper, all experiments were conducted on an NVIDIA RTX 3080 GPU with 12 GB of memory. In the process of training the CINet using WHRI, we set the number of training epochs to 90 and the batch size to 8. The sum of $l_{cls}$, $l_{obj}$, $l_{iou}$ was used as the quality indicator during training. The Adam optimizer [62] with default parameters was selected as the network optimizer. Furthermore, the learning rate was dynamically updated according to the number of training rounds (from $1 \times 10^{-3}$ to $1 \times 10^{-5}$).

The network was trained and inferred based on Pytorch. In the process of training the RoadBranchNet using MUNO21, we set the batch size to 4. The network's loss function is composed of three equal-weight components as in RoadTracer [45]: the action loss, angle loss, and cross-entropy loss between the predicted thumbnail and ground truth. The sum of the action loss, angle loss, and cross-entropy loss was used as the quality indicator during training. We used the Adam optimizer and trained 400 epochs. The initial learning rate was $1 \times 10^{-5}$ and was updated every 100 epochs. The network was trained and inferred based on TensorFlow.

#### 4.1.3. Comparative Methods

To evaluate the effectiveness of VecRoadUpd on road map updating, it was compared with six methods, including one semi-automatic road map update method called Maid [6] and five road extraction methods, including RoadConn [12] (road segmentation), Road-Tracer [45] (iterative road centerline tracing), Sat2Graph [46] (road centerline extraction), RecurrentUnet [22] (extract road surface and centerline simultaneously), and RNGDet [2] (road centerline extraction by transformer).

All methods were trained using the training set in MUNO21. We extended the road-extraction algorithm to road map updates using the fusion algorithm proposed by Bastani et al. [4]. The fusion algorithm fused the road extraction results with the old road map for road map updating. The comparison between our VecRoadUpd framework and the tested comparative methods on road map updates validated the efficiency of the proposed VecRoadUpd on vector road map updating.

#### 4.1.4. Evaluation Metrics

(1) Pixel-level metrics: To assess the improvement of the road update results in terms of completeness and correctness, we calculated the corresponding BaseMetric-improvement metric based on completeness (Comp), correctness (Corr), and quality (Qual) [63]. BaseMetric-improvement is used to measure the improvement of the Comp, Corr, and Qual metrics of the road update results compared to the corresponding metrics of the old road map. BaseMetric-improvement refers to Comp-improvement, Corr-improvement, and Qual-improvement. These are defined as follows:

$$Comp = \frac{length\ of\ matched\ reference}{length\ of\ reference} \tag{2}$$

$$Corr = \frac{length\ of\ matched\ extraction}{length\ of\ extraction} \tag{3}$$

$$Qual = \frac{length\ of\ matched\ extraction}{length\ of\ extraction + length\ of\ unmatched\ reference} \tag{4}$$

$$BaseMetric - improvement = \frac{BaseMetric(updated) - BaseMetric(old)}{1 - BaseMetric(old)} \tag{5}$$

where "updated" represents the updated road maps and "old" represents the old road maps.

(2) Graph-level metrics: To evaluate the improvement of the road update results in terms of topological correctness and connectivity, we use the precision and recall (the average path length similarity (APLS) [64] improvement) given in the MUNO21 dataset as evaluation metrics. To evaluate the precision, an error rate ($r_{error}$) is computed in each no-change scenario, which is used to indicate whether the map update method was executed correctly or not. If no change is inferred, i.e., the updated road map $\hat{G}$ = the pre-change map G = the post-change map $G^*$ (ground truth). The $r_{error}$ for that scenario is 0; otherwise, it is 1. The precision is defined as follows:

$$Precision = \frac{1}{N_{nc}} \sum_{i=0}^{N_{nc}} (1 - r(i)_{error}) \tag{6}$$

where $N_{nc}$ is the number of no-changed scenarios.

To evaluate the recall, the score is calculated using scenarios with changes. The score is used to indicate the degree to which $\hat{G}$ and $G^*$ are more similar than G and $G^*$. The recall is defined as follows:

$$Recall = \frac{1}{N_c} \sum_{i=0}^{N_c} \max \left( \frac{APLS(\hat{G}_i, G_i^*) - APLS(G_i, G_i^*)}{1 - APLS(G_i, G_i^*)}, -1 \right) \tag{7}$$

where $N_c$ is the number of changed scenarios.

In this paper, APLS is used to calculate the topological connectivity similarity between road map $G_1$ and road map $G_2$. APLS is defined as follows:

$$APLS(G_1, G_2) = \frac{1}{\frac{1}{S_{P \to T}(G_1, G_2)} + \frac{1}{S_{T \to P}(G_2, G_1)}} \tag{8}$$

$$S_{P \to T}(G_1, G_2) = 1 - \frac{1}{N} \sum \min\left(1, \frac{|Len(A_{G_1}, B_{G_1}) - Len(A_{G_2}, B_{G_2})|}{Len(A_{G_2}, B_{G_2})}\right) \tag{9}$$

where $N$ is the number of unique paths. The nodes $A_{G_2}$ and $B_{G_2}$ represent the nodes in the updated graph closest to the location of ground-truth nodes $A_{G_1}$ and $B_{G_1}$. The shortest path length of $A \to B$ in the ground truth is Len $(A_{G_1}, B_{G_1})$ and similarly Len $(A_{G_2}, B_{G_2})$. $S_{P \to T}$ measures the sum of the difference of the shortest path for each node pair in the ground-truth graph $G_1$ and the updated graph $G_2$.

Since the precision and recall evaluate the performance of the algorithm in updating road maps in unchanged and changed scenarios, they do not accurately evaluate the overall performance of the algorithm. Therefore, we introduce a reconciliation metric F1-score, which can reconcile the precision and recall to reflect the overall performance of different methods. The F1-score is calculated as follows:

$$F1\text{-}score = \frac{2 \times Precision \times Recall}{Precision + Recall} \tag{10}$$

### 4.2. Experimental Results

In this section, we present the visual and quantitative results of our VecRoadUpd compared to the other tested methods in road map updating. In the visualization results, due to the limited length of the article, we selected the road update results of three representative areas for display. San Antonio has clearer roads and less occlusion but with variable road materials. Washington DC has dense vegetation, an uneven distribution of buildings and roads, and variable road shapes. Los Angeles has dense buildings and a complex road network. In the display of quantitative results, we comprehensively evaluated the road update results of all tested methods on MUNO21 as shown in Section 4.2.4.

#### 4.2.1. Visual Results on San Antonio

The visual results in San Antonio effectively validated the performance of our VecRoadUpd for updating roads with similar material backgrounds and roads in simple scenarios. See Figure 7 for details. There are eleven columns in Figure 7. The subgraph (a) shown in the first column is an overview of the visual results of VecRoadUpd, where the updated road map is marked in yellow, and five regions are marked for zooming in. Columns two to eleven are the close-ups of the remote-sensing images, old road maps, ground truth, and visual results of all test methods in sequence.

As can be seen in Figure 7, in the simple scenario (the blue box and green box), the six tested methods and our VecRoadUpd have fewer missing and broken roads in the road update results. However, in the area where the road material is similar to the surrounding land (purple box), there are road breaks and and miss-detection in the results of the six comparative methods, while VecRoadUpd still keeps the roads connected and complete.

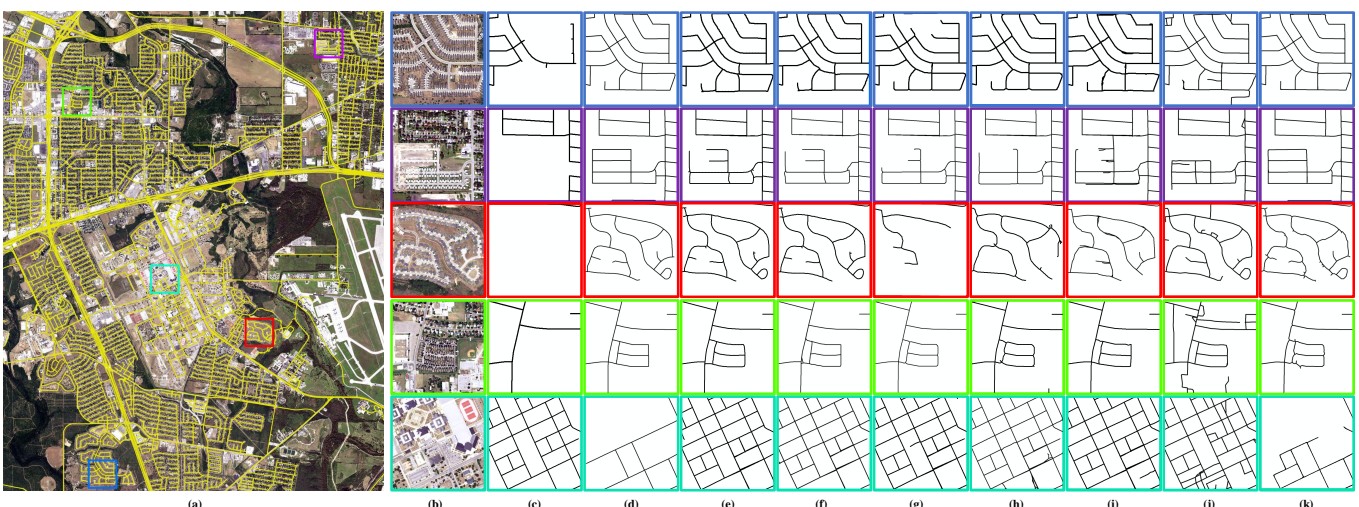

**Figure 7.** Visual results on San Antonio. (**a**) Overview result of the proposed VecRoadUpd. (**b**) Current image. (**c**) Old roads. (**d**) Ground truth. (**e**) Maid. (**f**) RoadTracer. (**g**) RecurrentUnet. (**h**) RoadConn. (**i**) Sat2graph. (**j**) RNGDet. (**k**) VecRoadUpd.

The reason is that VecRoadUpd is guided by the intersection branch to trace newly built roads, which is more concerned with the edge features of the road and less influenced by the road material. In areas with irregular road shapes marked with the red box, iterative methods, including Maid, RoadTracer, and VecRoadUpd, had fewer missed detections and road breaks compared with other methods. The reason is that iterative methods focus more on the connectivity and direction characteristics of the road. The road update results also show that our VecRoadUpd accurately deleted the disappeared roads in areas with disappeared roads (cyan box).

### 4.2.2. Visual Results on Washington DC

The visual results in Washington DC effectively validated the performance of our VecRoadUpd for updating road maps in areas with dense vegetation and variable road shapes. The details are shown in Figure 8.

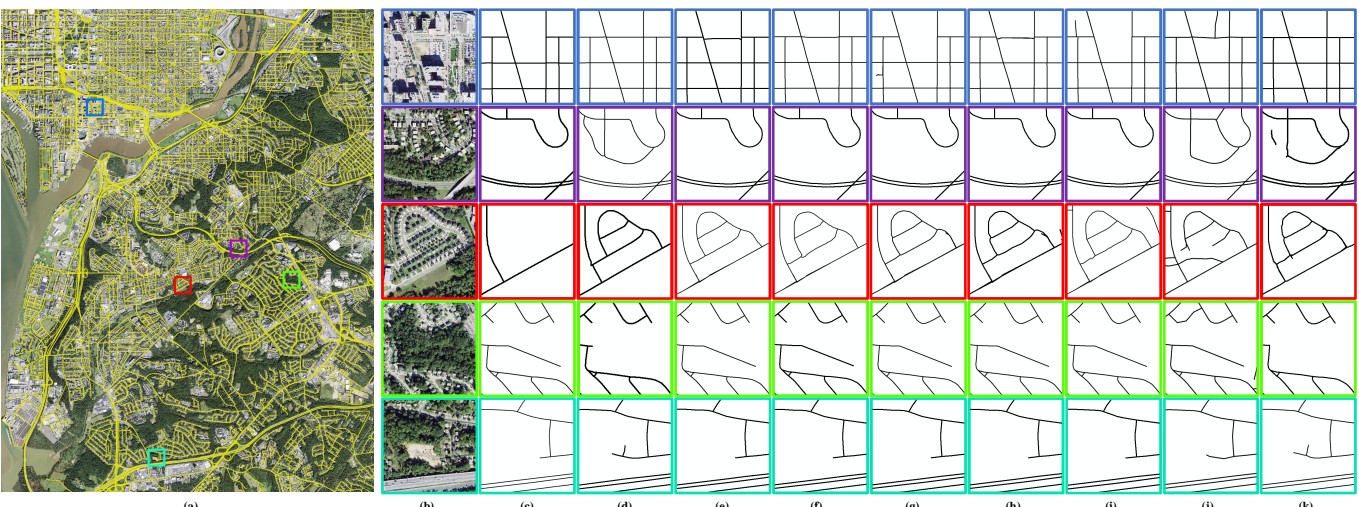

**Figure 8.** Visual results on Washington DC. (**a**) Overview result of the proposed VecRoadUpd. (**b**) Current image. (**c**) Old roads. (**d**) Ground truth. (**e**) Maid. (**f**) RoadTracer. (**g**) RecurrentUnet. (**h**) RoadConn. (**i**) Sat2graph. (**j**) RNGDet. (**k**) VecRoadUpd.

The overall results in Figure 8a show that VecRoadUpd ensured the integrity and connectivity of the updated road map in areas with dense vegetation and variable road shapes. The proposed VecRoadUpd and the other tested methods except RNGDet achieved perfect road update results in a simple scenario (red box) in areas shown by zooming in. In the scene with low contrast between the road and background (blue box), the iterative road tracing methods as well as RNGDet and RoadConn accurately updated the roads. However, in the similar scenario shown in the cyan box, only VecRoadUpd completely extracted the newly built roads.

We suggested that there are two reasons: (i) The road feature in this area is not obvious from the image, which makes it difficult for pixel-level road segmentation methods to detect newly built roads. (ii) The roads connected to newly built roads here are heavily obscured by vegetation and shadows, which causes the iterative road tracing method to be interrupted in the process of tracing roads. In contrast, the directed tracing in VecRoadUpd starts from the intersection detected in images, and thus the newly built roads are accurately extracted there. In the densely vegetated area shown in the purple box, the road is obscured by vegetation and shadows, while the road intersection is largely unobstructed.

Therefore, VecRoadUpd effectively overcomes the occlusion and accurately extracts parts of the newly built roads with the guidance of the intersections and their branches. The purple box also shows the importance of road intersections in the road tracing method. In addition, RNGDet also extracts the newly built roads on the right accurately. However, the newly built roads on the left of the purple box are not completely updated.

The miss-detected roads indicate that the current road tracing methods and transformer-based road graph extraction methods have not solved the problem of road occlusion by trees. The problem of road occlusion by trees is also a common problem for all road-extraction algorithms and needs to be addressed in future research. The road update results also show that our VecRoadUpd accurately deletes disappeared roads in areas with disappeared roads (green box).

### 4.2.3. Visual Results on Los Angeles

The visual results in Los Angeles effectively validated the performance of our VecRoadUpd for road updates in areas with dense buildings and complex road networks. The details are shown in Figure 9.

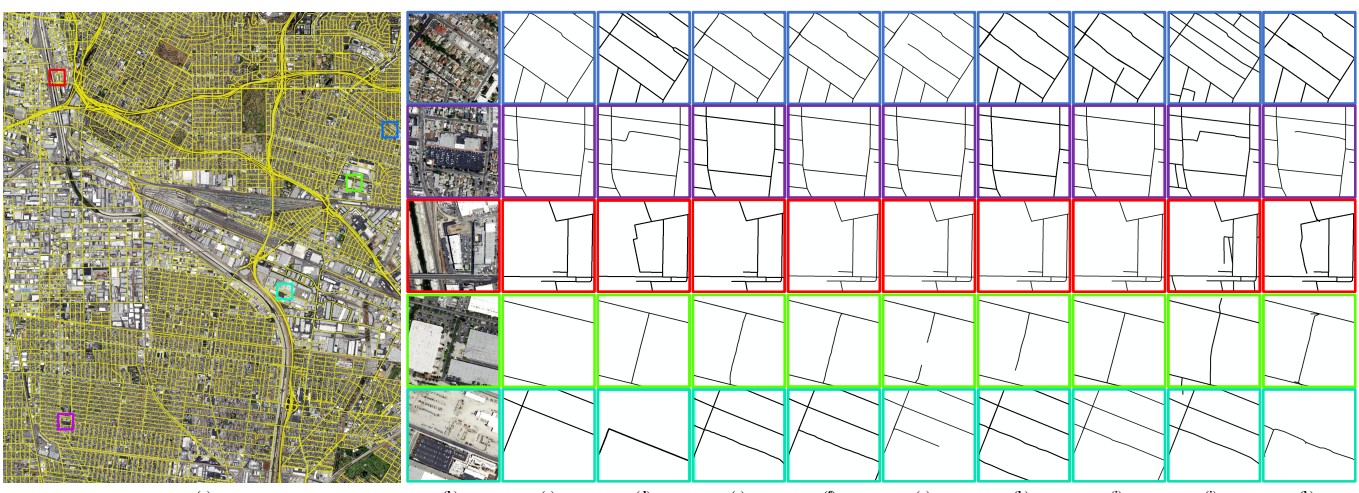

**Figure 9.** Visual results on Los Angeles. (**a**) Overview result of the proposed VecRoadUpd. (**b**) Current image. (**c**) Old roads. (**d**) Ground truth. (**e**) Maid. (**f**) RoadTracer. (**g**) RecurrentUnet. (**h**) RoadConn. (**i**) Sat2graph. (**j**) RNGDet. (**k**) VecRoadUpd.

It can be seen from Figure 9a that the proposed VecRoadUpd ensures the integrity and connectivity of the updated road map in areas with dense buildings and complex road networks. In the area shown by zooming in, all methods achieved high accuracy in

areas with distinguishable road characteristics and less occlusion, such as the newly built roads updated in the blue box, green box, and cyan box. In the area with complex road backgrounds (purple box), only RNGDet and VecRoadUpd extracted newly built roads, and RNGDet achieved better road connectivity, indicating that road tracing methods still require improvement in these similar areas.

In the area with variable road material (red box), all six comparative methods failed to extract the newly built roads. In contrast, the proposed VecRoadUpd extracted newly built roads accurately guided by newly built branches. The road update result obtained by VecRoadUpd further illustrates the key role of road intersections and branches in road updates.

### 4.2.4. Quantitative Analysis

Table 1 compares the quantitative results of the completeness and correctness of our VecRoadUpd with the six comparative methods for road map updates. The Comp-improvement, Corr-improvement, and Qual-improvement for each method on MUNO21 are shown in Table 1. The third to ninth rows show the individual metrics of the seven algorithms, and lines ten to fifteen show the differences between our proposed VecRoadUpd and the other methods.

**Table 1.** Quantitative pixel-level analysis of the road map update results.

| Methods | | Pixel-Level Metrics (%) [1] | | |
| --- | --- | --- | --- | --- |
| | | Comp-Improvement | Corr-Improvement | Qual-Improvement |
| The whole test cities | Maid [6] | 56.05 | 50.05 | 51.24 |
| | RecurrentUnet [22] | 55.97 | 50.36 | 51.34 |
| | RoadConn [12] | 60.92 | 39.63 | 49.01 |
| | RoadTracer [45] | 56.81 | 50.24 | 51.74 |
| | Sat2Graph [46] | 55.93 | 48.63 | 50.52 |
| | RNGDet [2] | 60.51 | 31.37 | 45.06 |
| | VecRoadUpd | **66.81** | **62.11** | **62.75** |
| Diff | VecRoadUpd-Maid | 10.76 | 12.06 | 11.51 |
| | VecRoadUpd-RecurrentUnet | 10.84 | 11.75 | 11.41 |
| | VecRoadUpd-RoadConn | 5.89 | 22.48 | 13.74 |
| | VecRoadUpd-RoadTracer | 10.00 | 11.87 | 11.01 |
| | VecRoadUpd-Sat2Graph | 10.88 | 13.48 | 12.23 |
| | VecRoadUpd-RNGDet | 6.30 | 30.74 | 17.69 |

[1] The highest evaluation scores are highlighted in bold. The second highest scores are marked with underlines.

As can be seen in Table 1, our VecRoadUpd achieved the highest scores in all pixel-level metrics, indicating that VecRoadUpd maintains the balance between the correctness and completeness of updated road maps. The difference section in Table 1 also shows that our VecRoadUpd improved 5.89%in Comp-improvement compared to the other tested methods. Combined with the visual results in Figures 7–9, VecRoadUpd had fewer omissions for newly built roads when compared with other tested methods.

For segmentation-based RoadConn and RecurrentUnet, high Comp-improvement and Corr-improvement scores were obtained due to the optimized pixel-level segmentation in the segmentation network. For the graph-based Maid, RoadTracer, Sat2Graphh, and RNGDet, they focus more on the topology of the road graph and the small roads in images, thus, yielding extra detections in unchanged regions and resulting in low Corr-improvement and Qual-improvement scores. Moreover, our method showed a significant improvement in Corr-improvement and Qual-improvement as compared to the other tested

methods, demonstrating that VecRoadUpd removes disappeared roads more accurately and rarely introduces errors for unchanged roads.

In addition, Table 2 compares the quantitative results on the topological correctness and connectivity of the updated road maps for all methods. The overall precision, recall, and F1-score of each method on MUNO21 as well as the recall and F1-score for each type of scenario are shown in the table. The fourth to tenth rows show the individual metrics for the seven algorithms, and lines eleven to sixteen show the differences between our VecRoadUpd and the six comparative methods.

**Table 2.** Quantitative graph-level analysis of the road map update results.

| Methods [1] | | Graph-Level Metrics (%) [2] | | | | | | | | | |
|---|---|---|---|---|---|---|---|---|---|---|---|
| | | | All | | Constructed | | Was-Missing | | Deconstructed | | Was-Incorrect | |
| | | Pre | Recall | F1 | Recall | F1 | Recall | F1 | Recall | F1 | Recall | F1 |
| The whole test scenarios | Ma [6] | **98.95** | 20.05 | 33.35 | 20.64 | 34.15 | 27.12 | 42.57 | <u>6.95</u> | <u>12.99</u> | 2.05 | 4.02 |
| | ReU [22] | 97.37 | 15.78 | 27.16 | 11.98 | 21.34 | 24.44 | 39.07 | 1.66 | 3.27 | 1.25 | 2.48 |
| | RC [12] | 75.26 | 15.25 | 25.36 | 16.47 | 27.03 | 23.47 | 35.78 | 0.88 | 1.75 | −0.67 | −1.36 |
| | RT [45] | 98.68 | 21.72 | 35.60 | 21.43 | 35.21 | <u>29.41</u> | <u>45.31</u> | 4.66 | 8.91 | <u>3.96</u> | <u>7.61</u> |
| | S2G [46] | 91.84 | 18.05 | 30.17 | 14.40 | 24.90 | 27.96 | 42.87 | 5.19 | 9.82 | −0.25 | −0.49 |
| | RNG [2] | 91.32 | <u>22.55</u> | <u>36.17</u> | <u>22.70</u> | <u>36.36</u> | 25.19 | 39.49 | 4.21 | 8.05 | 2.16 | 4.22 |
| | VecUpd | <u>98.70</u> | **33.50** | **50.02** | **28.10** | **43.75** | **36.79** | **53.60** | **21.38** | **35.15** | **27.78** | **43.36** |
| Diff | VecUpd-Ma | −0.25 | 13.45 | 16.67 | 7.46 | 9.60 | 9.67 | 11.03 | 14.43 | 22.16 | 25.73 | 39.34 |
| | VecUpd-ReU | 1.33 | 17.72 | 22.86 | 16.12 | 22.41 | 12.35 | 14.53 | 19.72 | 31.88 | 26.53 | 40.88 |
| | VecUpd-RC | 23.44 | 18.25 | 24.66 | 11.63 | 16.72 | 13.32 | 17.82 | 20.50 | 33.40 | 28.45 | 44.72 |
| | VecUpd-RT | 0.02 | 11.78 | 14.42 | 6.67 | 8.54 | 7.38 | 8.29 | 16.72 | 26.24 | 23.82 | 35.75 |
| | VecUpd-S2G | 6.86 | 15.45 | 19.85 | 13.70 | 18.85 | 8.83 | 10.73 | 16.19 | 25.33 | 28.03 | 43.85 |
| | VecUpd-RNG | 7.38 | 10.95 | 13.85 | 5.40 | 7.39 | 11.60 | 14.11 | 17.17 | 27.10 | 25.62 | 39.14 |

[1] Method names in the table are abbreviated due to table width limitations. Among them, Ma refers to Maid [6], ReU refers to RecurrentUnet [22], RC refers to RoadConn [12], RT refers to RoadTracer [45], S2G refers to Sat2Graph [46], RNG refers to RNGDet [2], and VecUpd refers to VecRoadUpd. [2] The highest evaluation scores are highlighted in bold. The second highest scores are marked with underlines.

As shown in Table 2, VecRoadUpd achieved the highest F1-score on all scenarios in MUNO21, which indicates that the VecRoadUpd updates changed roads accurately and maintained a low error rate in unchanged scenarios. For the Constructed and Was-missing scenarios, the recall of VecRoadUpd was improved by 5.4% for Constructed scenarios and 7.3% for Was-missing scenarios compared to RoadTracer. Higher recall values verify that the proposed directed tracing that updates roads with the guidance of changed road intersections extracted newly built roads more accurately and maintained the connectivity of the updated road map.

For segmentation-based RoadConn and RecurrentUnet, the road graph is extracted by skeletonization algorithms. These methods have high pixel-level scores as can be seen in Table 1. However, low graph-level scores were obtained since these methods cannot take full advantage of spatial and geometric information. For graph-based Maid, RoadTracer, Sat2Graphh, and RNGDet, the road graph is directly optimized in the network, thus, yielding higher graph-level scores in comparison with the segmentation-based approaches.

However, RoadTracer often fails to obtain high quality road maps when tracing to road intersections due to its fixed step size and limited number of starting points. Sat2Graph and RNGDet work well for unobstructed road detection but tend to produce more false detections in areas with high buildings or tree obstructions, such as dense urban areas, making the final performance degraded. For the Deconstructed and Was-incorrect scenes, all six comparative methods obtained low recall scores as they failed to remove disappeared roads.

However, VecRoadUpd achieved a recall score of 21.3% on Deconstructed scenes and 27.7% on Was-incorrect scenes, indicating that VecRoadUpd accurately removed roads from historical road maps that no longer exist in current images. Overall, our VecRoadUpd not only maintained high precision scores but also achieved a nearly 11% improvement in recall compared to the comparative methods, demonstrating that VecRoadUpd accurately updated the historical road map with almost no errors introduced.

## 5. Parameter Setting and Ablation Analysis

As mentioned in Section 3, the values of two parameters $T_{\mathrm{PIoU}}$ and $T_{\mathrm{angle}}$ in VecRoad-Upd affect the road map update, so this section analyzes the values of $T_{\mathrm{PIoU}}$ and $T_{\mathrm{angle}}$. $T_{\mathrm{PIoU}}$ is the threshold of PIoU in the road intersection change-detection rule. $T_{\mathrm{angle}}$ is the threshold of the absolute angle difference between the old and new branches in the road branch change-analysis rule. This section also analyzes the effectiveness of directed tracing in road map updates.

### 5.1. Influence of $T_{\mathrm{PIoU}}$

$T_{\mathrm{PIoU}}$ is used to detect changed road intersections. Different $T_{\mathrm{PIoU}}$s directly affect the detection results of changed intersections, which, in turn, affect the road map updates. In this section, the $T_{\mathrm{PIoU}}$ values were analyzed by implementing ablation experiments. In the experiments, the $T_{\mathrm{PIoU}}$ values varied from 0.6 to 0.85, and all other parameters in VecRoadUpd were kept constant. The detailed experimental results are shown in Table 3.

**Table 3.** Influence of $T_{\mathrm{PIoU}}$ on the performance of road map updating.

| $T_{\mathrm{PIoU}}$ | Metrics (%) [1] | | | | | | | | | | |
|---|---|---|---|---|---|---|---|---|---|---|---|
| | All | | | Constructed | | Was-Missing | | Deconstructed | | Was-Incorrect | |
| | Pre | Recall | F1 | Recall | F1 | Recall | F1 | Recall | F1 | Recall | F1 |
| 0.60 | **98.70** | 25.67 | 40.74 | 26.07 | 41.24 | 24.59 | 39.38 | 17.47 | 29.68 | 23.85 | 38.42 |
| 0.65 | **98.70** | 29.37 | 45.27 | 26.89 | 42.26 | 30.83 | 46.98 | 20.12 | 33.43 | 24.14 | 38.79 |
| 0.70 | **98.70** | **33.50** | **50.02** | **28.10** | **43.75** | **36.79** | **53.60** | **21.38** | **35.15** | **27.78** | **43.36** |
| 0.75 | 93.65 | 27.02 | 41.94 | 25.36 | 39.91 | 27.85 | 42.93 | 17.45 | 29.41 | 23.14 | 37.10 |
| 0.80 | 90.21 | 26.33 | 40.76 | 25.39 | 39.62 | 26.76 | 41.28 | 17.25 | 28.96 | 20.92 | 33.97 |
| 0.85 | 75.22 | 24.96 | 37.49 | 23.97 | 36.35 | 27.12 | 39.86 | 13.53 | 22.93 | 16.72 | 27.36 |

[1] The highest evaluation scores are highlighted in bold.

Table 3 shows that, when $T_{\mathrm{PIoU}}$ was set to 0.7, VecRoadUpd achieved the highest score out of all values. When the $T_{\mathrm{PIoU}}$ was less than 0.75, the precision of VecRoadUpd remained at 0.9870; however, the recall and F1-score decreased. When the $T_{\mathrm{PIoU}}$ value is small, the overlap between old and new intersections is high in unchanged scenarios, making PIoU $\geq T_{\mathrm{PIoU}}$ identify the intersection as an unchanged intersection. Therefore, VecRoadUpd keeps the error rate low in unchanged scenarios to obtain high precision. In the changed scenarios (Constructed, Was-missing, Deconstructed, and Was-incorrect), low $T_{\mathrm{PIoU}}$ leads to changed intersections miss, resulting in a low recall and F1-score.

Table 3 also shows that, when the $T_{\mathrm{PIoU}}$ value was greater than 0.7, the precision, recall, and F1-score decreased as the $T_{\mathrm{PIoU}}$ increased. The reason is that, in the unchanged scenarios, although the overlap between old and new intersections is high, the $T_{\mathrm{PIoU}}$ is higher, which causes some unchanged intersections to be misclassified as changed intersections. Even though such errors are likely to be corrected during branch change detection, the probability of correction is low. Therefore, it falsely detects the changed roads, resulting in errors in unchanged scenarios and a decrease in precision. In the changed scenarios, the same reason leads to false detection of the changed roads, decreasing both the recall and F1-score.

## 5.2. Influence of $T_{\text{angle}}$

$T_{\text{angle}}$ is used to decide whether road branches are changed. Different $T_{\text{angle}}$s directly affect the detection results of changed branches, which, in turn, affect the road map update results. In this section, the $T_{\text{angle}}$ values were analyzed by implementing ablation experiments. In the experiments, the $T_{\text{angle}}$ values varied from $\pi/32$ to $3\pi/16$. The detailed experimental results are shown in Table 4.

Table 4 shows that, when $T_{\text{angle}}$ was set to $\pi/8$, VecRoadUpd achieved the highest score out of all values. When the $T_{\text{angle}}$ was greater than $3\pi/32$, the precision of VecRoadUpd stayed at 0.9870; however, the recall and F1-score decreased as the $T_{\text{angle}}$ increased. When $T_{\text{angle}}$ is large, in unchanged scenes, the angle difference is smaller than $T_{\text{angle}}$ due to the high overlap between the old and new branch directions, and the branch is considered an unchanged branch. Therefore, VecRoadUpd keeps the error rate low in unchanged scenarios to obtain high precision.

In the changed scenarios, high $T_{\text{angle}}$ leads to changed branches being missed, resulting in a low recall and F1-score. It can also be seen from Table 4 that, when the value of $T_{\text{angle}}$ was less than $\pi/8$, the precision, recall, and F1-scores decreased as the $T_{\text{angle}}$ decreased. In the unchanged scenarios, although the difference between old and new branch directions is low, the $T_{\text{angle}}$ is lower, which causes some unchanged branches to be misclassified as direction-changed branches. In the changed scenarios, the same reason leads to false detection of the changed roads, decreasing the recall and F1-score.

**Table 4.** Influence of the $T_{\text{angle}}$ on the performance of road map updating.

| $T_{\text{angle}}$ | Metrics (%) [1] | | | | | | | | | | |
|---|---|---|---|---|---|---|---|---|---|---|---|
| | All | | | Constructed | | Was-Missing | | Deconstructed | | Was-Incorrect | |
| | Pre | Recall | F1 | Recall | F1 | Recall | F1 | Recall | F1 | Recall | F1 |
| $\pi/32$ | 63.26 | 25.68 | 36.53 | 24.96 | 35.80 | 25.73 | 36.58 | 17.83 | 27.82 | 22.79 | 33.50 |
| $\pi/16$ | 89.31 | 26.71 | 41.12 | 27.36 | 41.88 | 27.30 | 41.82 | 11.94 | 21.06 | 16.27 | 27.53 |
| $3\pi/32$ | 93.45 | 29.48 | 44.82 | 27.58 | 42.59 | 30.13 | 45.57 | **23.53** | **37.60** | 23.45 | 37.49 |
| $\pi/8$ | **98.70** | **33.50** | **50.02** | **28.10** | **43.75** | **36.79** | **53.60** | 21.38 | 35.15 | **27.78** | **43.36** |
| $5\pi/32$ | **98.70** | 31.78 | 48.08 | 26.57 | 41.87 | 35.64 | 52.37 | 8.15 | 15.07 | 24.53 | 39.29 |
| $3\pi/16$ | **98.70** | 28.62 | 44.37 | 23.88 | 38.46 | 31.86 | 48.17 | 20.62 | 34.11 | 21.43 | 35.22 |

[1] The highest evaluation scores are highlighted in bold.

## 5.3. Influence of Directed Road Tracing

In VecRoadUpd, directed tracing was developed to extract newly built roads and verify disappeared roads. To evaluate the impact of directed tracing on road map updates, we replaced it with MspTracer and conducted experiments on MUNO21. The starting points in MspTracer are intersections detected by CINet in images. To improve the efficiency of MspTracer, we popped the intersections already traced from the starting points stack during the tracing process to avoid repeated tracing. The $T_{\text{PIoU}}$ used in the experiment was 0.7, and the $T_{\text{angle}}$ was $\pi/8$. Four representative visual results are shown in Figure 10.

In Figure 10, we selected Constructed and Was-missing scenarios containing the newly built roads for visualization. It can be seen from Figure 10 that the newly built road extraction results of the two algorithms are the same. However, MspTracer cannot accurately delete the disappeared roads, which is the main reason why MspTracer is lower than directed tracing in all quantitative metrics. The comparison of graph-level metrics for the two algorithms is shown in Table 5.

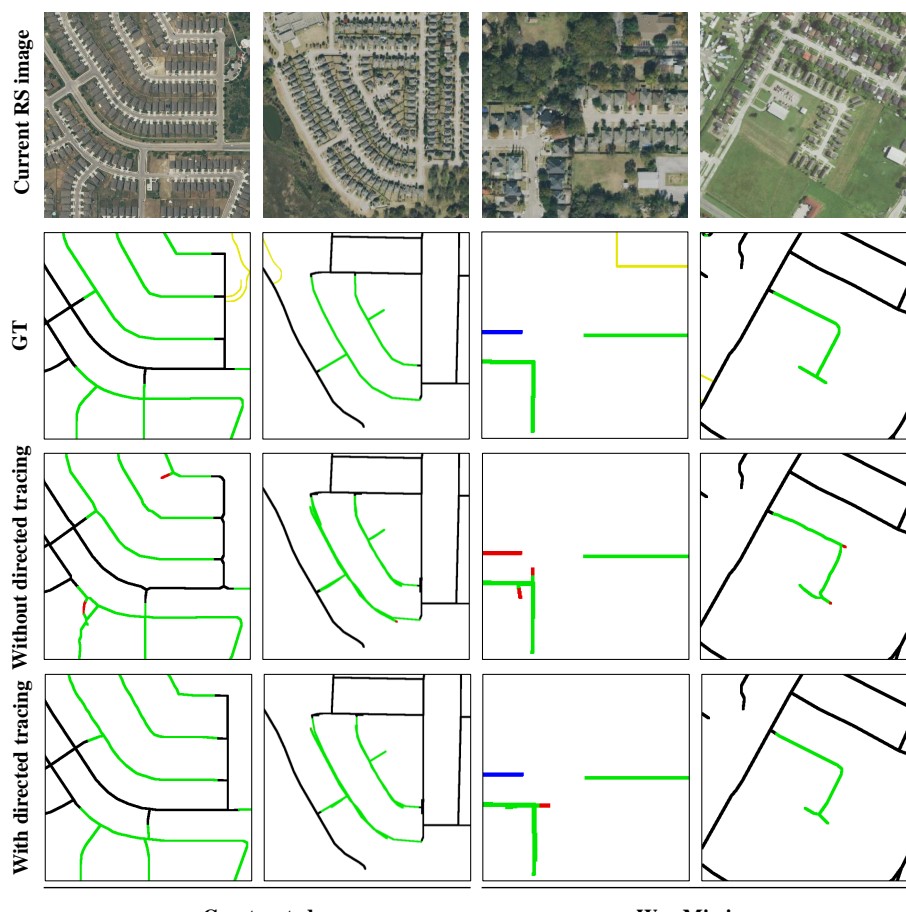

In the figure, added roads are shown in green(correct) or red (wrong), removed roads are shown in blue (correct) or orange (wrong), unchanged roads are shown in black, and footpaths are shown in yellow.

**Figure 10.** Illustration of the results on MUNO21 with and without directed tracing.

**Table 5.** Quantitative graph-level analysis of the road map update results with or without directed tracing.

| Methods | Graph-Level Metrics (%) | | | | | | | | | |
|---|---|---|---|---|---|---|---|---|---|---|
| | All | | | Constructed | | Was-Missing | | Deconstructed | | Was-Incorrect | |
| | Pre | Recall | F1 | Recall | F1 | Recall | F1 | Recall | F1 | Recall | F1 |
| with Directed_Tracing | 98.70 | 33.50 | 50.02 | 28.10 | 43.75 | 36.79 | 53.60 | 21.38 | 35.15 | 27.78 | 43.36 |
| without Directed_Tracing | 98.67 | 28.05 | 43.68 | 27.06 | 42.48 | 36.03 | 52.79 | 7.67 | 14.24 | 8.44 | 15.55 |

From Table 5, it can be seen that the graph-level metrics of these two algorithms are basically the same in the Constructed and Was-missing scenarios. However, directed tracing improved the overall recall and F1 scores by about 5.5% and 6.5%. The reason is that directed tracing had a high recall and F1 score in the Deconstructed and Was-incorrect scenarios. The precision of the two algorithms is almost equal; however, directed tracing had a 0.3% improvement. The 0.3% improvement indicates that the branch change detection makes a small correction to the intersection change detection results. In addition, the comparison of the two algorithms in terms of pixel-level metrics and time spent is shown in Table 6.

**Table 6.** Quantitative pixel-level analysis of the road map update results with or without directed tracing.

| Methods | Pixel-Level Metrics (%) | | | Inference Time (h) |
|---|---|---|---|---|
| | Comp-Improvement | Corr-Improvement | Qual-Improvement | |
| with Directed_Tracing | 66.81 | 62.11 | 62.75 | 2.1347 |
| without Directed_Tracing | 66.82 | 55.01 | 59.61 | 8.9538 |

From Table 6, it can be seen that the application of directed tracing brings significant improvement in the correctness metrics and quality metrics of the updated road map, and the time consumption is significantly reduced. The Comp-improvement metric of MspTracer is slightly higher than that of directed tracing by 0.1%. The high of 0.1% is because MspTracer cannot remove disappeared roads, resulting in some of the disappeared road pixels that overlap with the ground truth being considered correct. However, the Corr-improvement and Qual-improvement metrics of directed tracing are significantly higher than those of MspTracer, owing to MspTracer's inability to accurately remove disappeared roads and resulting in too many false detections. The last column of Table 6 also shows that it took about nine hours for MspTracer to update the road maps in MUNO21, while directed tracing took only about two hours. This indicates that our directed tracing takes less time and is more efficient.

## 6. Discussion on Failure Cases

Some failure cases of the proposed VecRoadUpd framework are visualized in Figure 11. For the first failure case, VecRoadUpd incorrectly deleted a portion of the non-disappeared road due to tree occlusion, resulting in a broken road. For the second failure case, also due to tree occlusion, VecRoadUpd did not update the road map accurately. Furthermore, the road map update results of the other methods in Figure 11 also show that tree occlusion caused all methods to fail to update the road map accurately. Tree occlusion was the most common cause of failure in road extraction and updating, which is useful information for improving our VecRoadUpd.

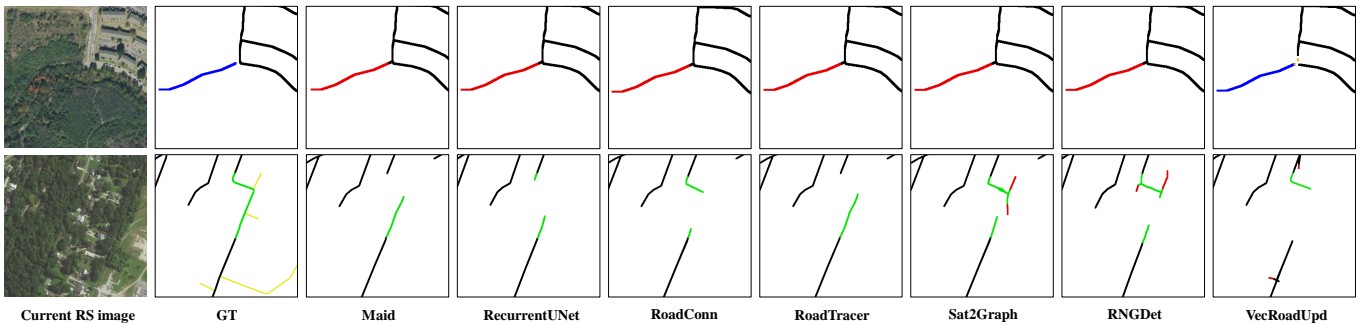

| | | | | | | | |
|---|---|---|---|---|---|---|---|
| **Current RS image** | **GT** | **Maid** | **RecurrentUNet** | **RoadConn** | **RoadTracer** | **Sat2Graph** | **RNGDet** | **VecRoadUpd** |

In the figure, added roads are shown in green(correct) or red (wrong), removed roads are shown in blue (correct) or orange (wrong), unchanged roads are shown in black, and footpaths are shown in yellow.

**Figure 11.** Some failure cases of the proposed VecRoadUpd framework.

## 7. Conclusions

In this paper, a vector road map updating framework (VecRoadUpd) was proposed for updating historical vector road maps based on changed road intersections instead of roads extracted from scratch as in the existing road updating methods.

The VecRoadUpd framework takes current images and historical road maps as the input and outputs updated road maps. The VecRoadUpd framework discovers and updates historical vector road maps using the change detection first and update later strategy. First, VecRoadUpd extracts intersections from current images and historical road maps using a CNN-based intersection-detection network (CINet). Then, VecRoadUpd identifies changed intersections based on the road intersection change-detection rule. Based on the discovery

of changed intersections, VecRoadUpd detects intersection branches from current images and old road maps using the road branch detection network (RoadBranchNet).

Then, VecRoadUpd identifies direction-changed road branches based on the road branch change-detection rule. After road map change discovery, a CNN-based directed tracing algorithm was introduced to extract and verify the changed roads for accurate road map updating. The algorithm starts tracing from the center points of changed intersections, and the tracing directions are restricted by direction-changed road branches to accurately extract and verify the changed roads. Finally, updated road maps were obtained.

Extensive experiments on MUNO21, a large road map update dataset containing 21 cities and 1294 different scenarios, confirmed the effectiveness of the proposed VecRoadUpd in the road map update task and also showed that road intersections play an important role in road map change discovery. However, VecRoadUpd did not update historical vector road maps accurately in areas with severe tree occlusion. The problem of tree occlusion has always been a difficulty in road extraction in complex scenes. Therefore, we will continue to investigate how to accurately update road maps in complex road scenarios in the future.

**Author Contributions:** Conceptualization, H.S., N.Z. and M.Z.; methodology, N.Z. and M.Z.; software, N.Z.; validation, N.Z. and M.Z.; formal analysis, N.Z.; investigation, N.Z., M.Z. and L.G.; resources, H.S.; data curation, H.S.; writing—original draft preparation, N.Z.; writing—review and editing, M.Z., H.S. and N.Z.; visualization, N.Z.; supervision, N.Z.; project administration, N.Z.; funding acquisition, H.S. All authors have read and agreed to the published version of the manuscript.

**Funding:** This research was funded by Guangxi Science and Technology Major Project (AA22068072).

**Data Availability Statement:** Publicly available datasets were analyzed in this study. The MUNO21 dataset can be found here: (https://favyen.com/muno21/ (accessed on 11 October 2021), MUNO21: A Dataset for Map Update using Aerial Images). The WHRI dataset can be found here: (http://www.lmars.whu.edu.cn/suihaigang/index.html (accessed on 1 June 2022), WHRI: WuHan Road Intersection Dataset).

**Acknowledgments:** The authors thank the teams of the datasets and algorithms used in this work. Our deepest gratitude goes to the reviewers and editors for their careful work and thoughtful suggestions that have helped improve this paper substantially.

**Conflicts of Interest:** The authors declare no conflict of interest.

## Appendix A

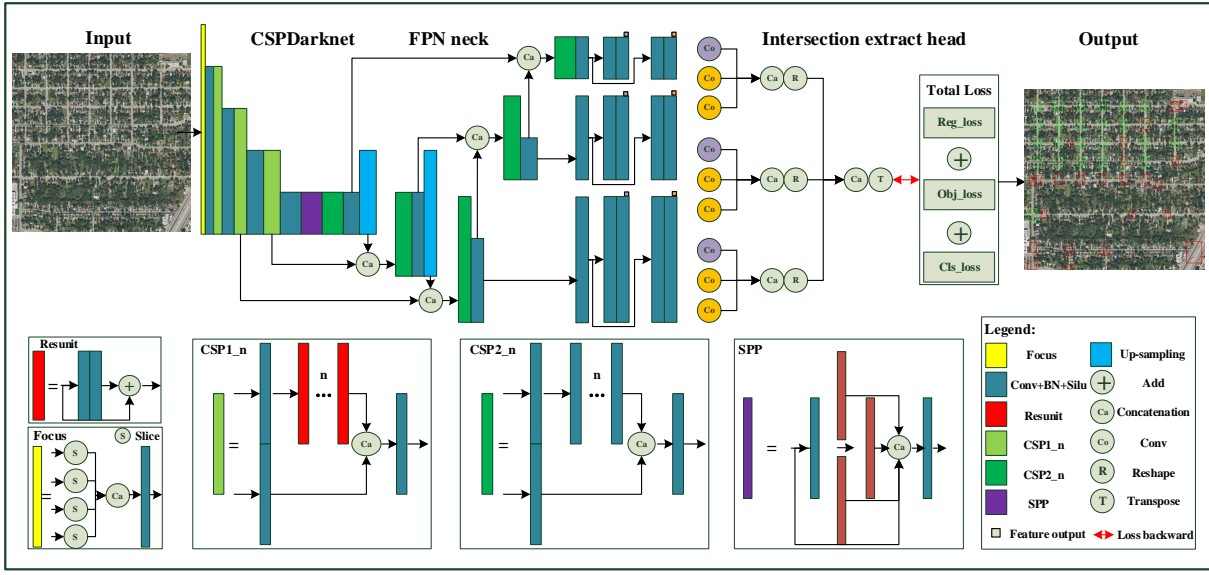

**Figure A1.** The architecture of the CNN-based intersection-detection network (CINet).

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
