# Peer review of "Vector Road Map Updating from High-Resolution Remote-Sensing Images with the Guidance of Road Intersection Change Detection and Directed Road Tracing"

_remotesensing, doi:10.3390/rs15071840_

Round 1

Reviewer 1 Report

This work is good and attractive. It gives a novel pipeline for vector road map updating from HR-RSI. I would like to see this paper published in Remote Sensing soon. Here, I have some suggestions and questions.

1. What is the time listed in Table 6? How long does the training phase consume? And, what is the inference speed of your model?

2. Please release your source code if possible. As there are many branches in your framework, a deep understanding is not easy to get.

3. Please verify the link given in Data Availability Statement. I have not found related data there.

4. Your method is closely related to change detection. For completeness, the related work part should include the recent advances in change detection, e.g., ISNet: towards improving separability for remote sensing image change detection (TGRS2022), BIT (TGRS2022), etc.

5. Please update your figures with high-quality ones. If you are Visio users, I have a suggestion for you. (File -> save as -> *.jpg -> save -> Quality: 100%, Resolution: printer)

6. What is ‘burred road centerline extraction’? Is it a typo (blurred)?

7. Please carefully check your symbols and notations. ‘max’ instead of ‘MAX’ or ‘max’. ‘angle’ instead of ‘angle’ as the subscript. Use mathematical symbols instead of *, ABS(), and something like that. 

Reviewer 2 Report

  In this paper, the authors propose a method for road updating. The methodology is clear and understandable. Here are two comments on the experiment:

1) The author should focus on analyzing the impact of the differences between methods  rather than describing the numerical differences in section 4.2.4.

2) Some recent methods should be added as comparison methods instead of using only the methods in MUNO21.

Reviewer 3 Report

Road map updating involves verifying unchanged roads, extracting newly-built roads, and removing disappeared roads. In this manuscript, a novel road vector map updating (VecRoadUpd) framework is proposed based on the observation that road changes are highly correlated with road intersection changes. But there

Q1. In the Figure 5. Flowchart of road branch change detection, should the information from Unchanged road branches be used to updated road map?

Q2. It is suggested that the  available datasets in this study should be shared by public cloud storage.

Reviewer 4 Report

The paper entitled "Vector Road Map Updating from High-Resolution Remote Sensing Images with the Guidance of Road Intersection Change Detection and Directed Road Tracing" proposes a novel framework (VecRoadUpd) for detecting changes in road intersections from new aerial imagery and available mapping to update road mapping where changes are often required or found. The framework consists of two parts, the first being a network (CINet) trained to detect road junction locations in the imagery and another network (RoadBranchNet) trained to detect the direction of road branches for each road junction to find road branch changes.

In related work, we move from segmentation to centreline extraction. In the proposed bibliography there are references to general segmentation issues, but no reference is made to semantic segmentation of roads or road axes using CNN. The state of the art in these aspects should be complemented.

The methodological approach is well described and argued. However, neither CINet nor RoadBranchNet are described correctly. The former is an object identification network specialised in detecting junctions. The network is not described, it is only indicated that it is based on CSPDarkNet53 , FPN and decoupled head. The proposed backbone is part of the known YOLOv4 network. This sequencing is the YOLOv4 network. Regarding the RoadBranchNet network, it is indicated that it is inspired by the CNN-based decision module designed in RoadTracer, but it is not described. In Figure 4 (a) one of the outputs is the semantic segmentation (of the first layers) Oseg, the others are Oact and Oang. This network is not described. Nor is it indicated how the angle (Oang) is calculated or was it part of the training dataset and the RoadBranchNet provides an array of angles as a result?

Both networks have at some point had to be trained with a dataset, of which some reference is made based on MUNO21, but there is also talk of the manually labelled WHRI dataset but no description of such a dataset is given (how many images form it, the division for training, etc...).

There is no justification in the first case for the choice of this network or architecture or for the choice of a well-established one such as Yolov4, or other versions of YOLO.

No quality indicators are provided for the training of the networks with the training dataset. However, much emphasis is then placed on benchmarking by comparing the results of the proposed framework with those existing in the reviewed literature.

Just as a link to a website is provided to access the dataset used in the experiments, the authors are invited to share the construction code of the networks used (CINet and RoadBranchNet), so that they have been able to benchmark against other methodologies, new researchers can compare their proposals with theirs.

Minor aspects of the review that can be highlighted:

Line 190 the acronym BCE loss appears without saying what Binay Cross Entropy defines, it only appears once the acronym may not be used. There also seems to be a typo in CIOU[41], not DIOU (Distance IoU)?

Round 2

Reviewer 4 Report

The authors have addressed all the recommendations made in the first review, including the improvement of the state of the art, the description of the networks used for the workflow and the explanations given to the comments are correct.

For this reason I consider that the article can be accepted for publication.